# Rate-Limiting Steps of Dye Degradation over Titania-Silica Core-Shell Photocatalysts

**Ariane Giesriegl**[ID]**, Jakob Blaschke, Shaghayegh Naghdi**[ID] **and Dominik Eder ***[ID]

Technische Universität Wien, Institut für Materialchemie, Getreidemarkt 9, 1060 Vienna, Austria
* Correspondence: dominik.eder@tuwien.ac.at

**Abstract:** In this work, we employed a step-by-step sol-gel process to controllably deposit ultra-thin layers of $SiO_2$ on anatase nanoparticles in the range between 0 and 1 nm. The deposition was confirmed by TEM, EDX, and ATR-FTIR (e.g., Ti-O-Si band at 960 cm$^{-1}$). Zeta potential measurements unravelled a continuous change in surface charge density with increasing silica shell thickness. The photocatalysts were evaluated towards adsorption and degradation of positively-charged and negatively-charged dyes (methylene blue, methyl orange) under UV illumination. The growth mechanism follows the Stranski–Krastanov model with three thickness regimes: (a) Flat islands (first step), (b) mono/bilayers (second/third step), and (c) regular thick films (fourth/fifth step). The results suggest different rate limiting processes for these regimes: (a) For the thinnest scenario, acidic triple-phase boundaries (TPBs) increase the activity for both dyes with their accessibility being the rate limiting step; (b) for continuous mono/bilayers, dye adsorption on the negatively-charged $SiO_2$ shells becomes the rate liming step, which leads to a stark increase in activity for the positively-charged MB and a decrease for MO; (c) for thicker shells, the activity decreases for both dyes and is limited by the charge transport through the isolating shells.

**Keywords:** dye degradation; core-shell; photocatalyst; titania; reaction mechanism

## 1. Introduction

$TiO_2$ is a widely-used and common reference photocatalyst as a model system for fundamental and applied photocatalytic studies. It is popular in photocatalysis because of its low cost, nontoxicity, and photostability [1,2], as well as its broad spectrum of applications as self-cleaning films, water splitting reactions, $CO_2$ reductions, and several mineralisations of environmental pollutions in aqueous media [1–5]. The efficiency of heterogeneous catalysts in general depends on the accessible surface area, crystallinity, and morphology. The small particle size in the nanometer region and anatase as the most photoactive crystalline phase of titania both lead to more photocatalytic efficiency [6]. There are several problems associated with titania, especially as nanoparticles. These include the recombination of photogenerated electron-hole pairs [5] and the necessity of UV light to generate electron-hole pairs due to its wide band gap ($E_g \approx 3.2$ eV) [7,8]. While the agglomeration of nanoparticles often causes deactivation, nanocatalysts are also harder to recover from suspensions [2,6].

Core-shell structures can offer a tunable model system for functional and mechanistic investigations in heterogeneous catalysis. Typically, core-shell materials are inorganic and most often include $SiO_2$ [9,10]. As an inert material, it is either used as support for a catalytic shell or as an insulating shell around a catalyst [2,5,6,11–13]. As observed in the literature, the coating of semiconductor photocatalysts with $SiO_2$ can have several advantages. It can lead to the reduced recombination of electron-hole pairs, tunable adsorption of reactants, and less agglomeration of nanoparticles [2,5,13–15]. However, as an insulating layer, it will inhibit charge transfer and will shield the photocatalytically-active surface from reactants [16,17]. It remains a challenge to control

the deposition of uniform ultra-thin shells on core nanoparticle catalysts with such a low thickness, in particular below 1 nm [10]. A common synthesis technique for metal oxide core-shell materials involves the adsorption of metal alkoxide precursors onto a nanoparticle surface, followed by hydrolysis and condensation reactions, and calcination procedures. $TiO_2 - SiO_2$ core-shell structures have been investigated in the wider context of photocatalysis, but the shells have mostly been limited to thick (several nm) and/or mesoporous coatings [6,13,18–21]. One of the observations was that coatings of several nm thickness typically diminish the photocatalytic activity [22,23]. So far, only a few groups have investigated the effects of a coating thinner than 2 nm, often with irregular thickness and morphology, or without comparing the photocatalytic effects on differently charged reactants [17,24–26]. In order to unambiguously investigate the effects of $SiO_2$ on photocatalytic properties of $TiO_2$, including effects on adsorption and charge transfer, weexpectto need non-porous $SiO_2$ layers thinner than 1 nm.

Therefore, in this work we prepared continuous films with a uniform thickness adjusted between 0 and 1 nm. We compared, for the first time, the photocatalytic effects on differently charged dyes in this crucial thickness range. This thickness range has allowed to unravel charge transfer limitations, reveal differences in adsorption behaviour, and accessibility to acidic sites and thus to identify rate limiting steps for the respective thickness regimes, which is crucial for the community to further advance the design of photocatalysts.

## 2. Results and Discussion

### 2.1. Physical Properties of Core-Shell $TiO_2 - SiO_2$ Nanoparticles

#### 2.1.1. Shell Thickness and Surface Properties

In this work, we synthesised core-shell structures by employing a layer-by-layer process that involves a step-wise impregnation, catalytic hydrolysis, and condensation of restricted amounts of tetraethyl orthosilicate (TEOS) on anatase nanoparticles and investigated film thickness and quality after steps 1 to 5 (TS1-5). The experimental details are summarised in the Methods section.

The shell structure and thickness was studied by TEM (Figure 1). The images show that the shells of steps 1 and 2 are too thin to be seen at the resolution of the current setup. EDX analysis, however, confirms the presence of Si (Table 1). The shells of steps 3–5 are clearly visible and growing in thickness with the increase of the number of steps.

An amorphous layer is visible for 3–5 steps of coating. The layer for TS3 is patchy and inhomogeneous and becomes more homogeneous with increasing thickness. This layer growth behaviour can be explained with the Stranski–Krastanov growth model, which states that adatoms that form the first monolayer attach preferentially on the supports surface until it is covered. This is theoretically more likely than the Volmer–Weber growth, which states that adatom-adatom interactions are stronger than those of the adatoms and the surface atoms, due to the lower electronegativity of Ti compared to Si, which leads to a faster condensation of the silica precursor on $TiO_2$ than with itself [27]. After the first monolayer coverage, an island formation is possible, leading to a patchy looking layer which becomes more even with increasing layer thickness. Table 1 shows the average thickness from at least 15 measurements evaluated by TEM.

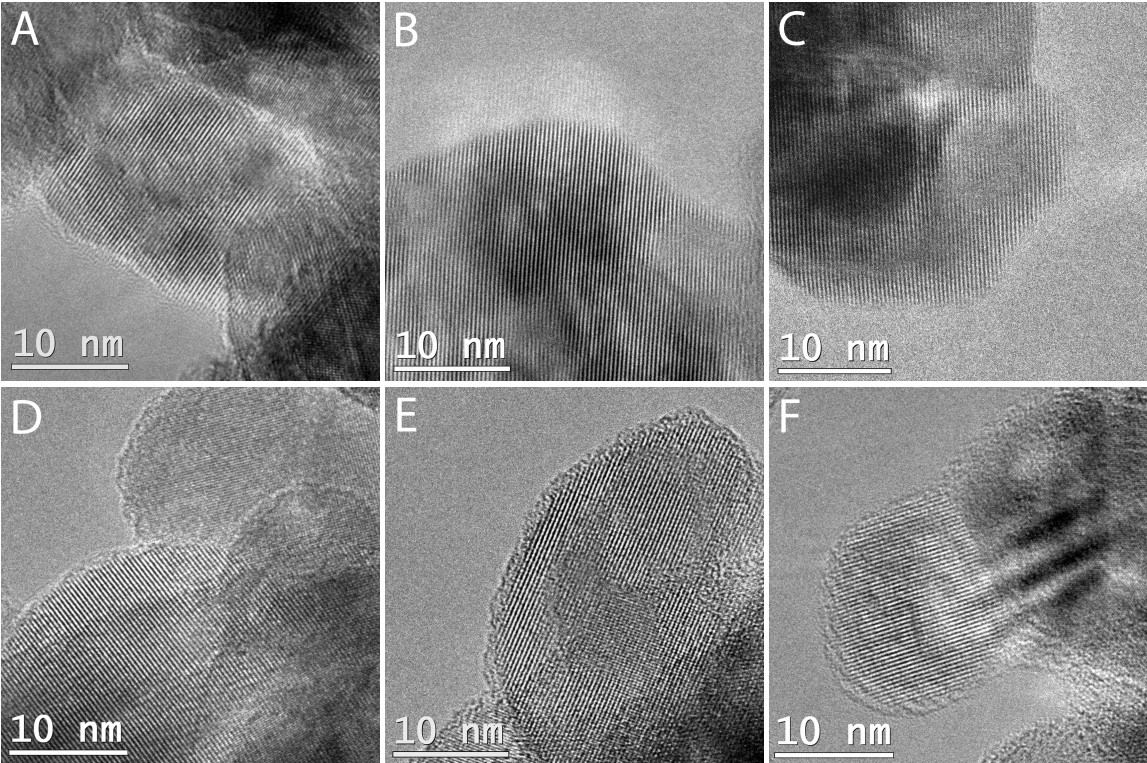

**Figure 1.** Typical transmission electron microscopy (TEM) images of samples (**A**) TS0, (**B**) TS1, (**C**) TS2, (**D**) TS3, (**E**) TS4, and (**F**) TS5.

**Table 1.** Theoretical Si:Ti ratio, $SiO_2$ shell thicknesses measured by TEM and EDX, specific surface areas (SSA), t-plot external area and t-plot micropore area from BET.

| Sample | Si:Ti [%] | TEM [nm] | EDX [nm] | $SSA_{BET}$ [$m^2g^{-1}$] | $SSA_{ext.}$ [$m^2g^{-1}$] | $SSA_{t-plot}$ [$m^2g^{-1}$] |
|---|---|---|---|---|---|---|
| TS0 | 0 | - | $0.03 \pm 0.03$ | 92.6 | 92.9 | - |
| TS1 | 4.3 | - | $0.1 \pm 0.02$ | 88.5 | 86.6 | 1.9 |
| TS2 | 8.6 | - | $0.29 \pm 0.02$ | 80.1 | 68.4 | 11.7 |
| TS3 | 12.9 | $0.63 \pm 0.13$ | $0.61 \pm 0.03$ | 81.2 | 70.6 | 10.5 |
| TS4 | 16.8 | $0.84 \pm 0.18$ | $0.81 \pm 0.04$ | 54.5 | 43.0 | 11.5 |
| TS5 | 21.4 | $0.97 \pm 0.13$ | $0.91 \pm 0.09$ | 74.4 | 52.1 | 22.3 |

The EDX results show (a) absence of any foreign impurities (Figure 2a), (b) Si is well distributed over the entire sample, and (c) Si content increases with the increase of steps. Assuming a homogeneous coverage, round particles, and a $SiO_2$ density of 2 g/cm$^3$, the calculated thicknesses taken from the respective Si:Ti ratio correlate well with those determined by TEM (Table 1). This leads to the conclusion that TS2 has a silica monolayer of (0.31 nm) as a shell, while TS1 has islands of silica on approximately 1/3 of the surface [25,28].

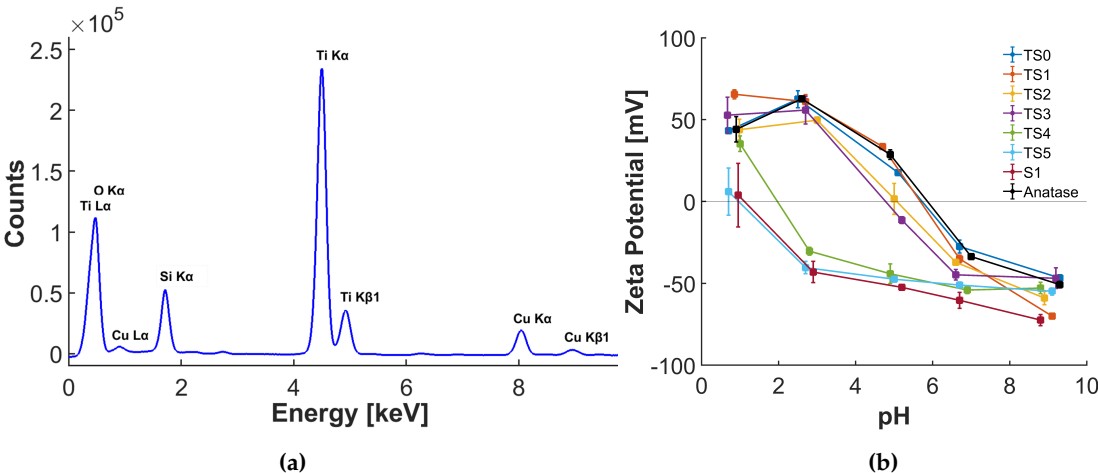

**Figure 2.** (**a**) Energy-dispersive X-ray spectrum (EDX) of the TS4 sample. (**b**) Variations of zeta potentials with pH of reference anatase, synthesised silica, and core-shell samples with various step numbers.

Table 1 shows changes of the specific surface area (SSA) for the various synthesis steps. The total SSA according toBET(SSA$_{BET}$) decreases slightly with the increasing number of steps. Analysis from the t-plot reveals that part of the SSA (SSA$_{t-plot}$) is composed of the inner surface area within micropores (pores with a diameter < 2 nm) [17]. The presence of micropores in the amorphous silica shells is inherent to low-temperature sol-gel processes. Therefore, it is plausible that the contribution of micropores to the total surface area increases with each increasing step number. We calculated the difference between the SSA$_{BET}$ and SSA$_{t-plot}$ as the active, external surface area (SSA$_{ext.}$). SSA$_{ext.}$ decreases considerably with each increasing step number, which we attributed to the increasing tendency of these core-shell structures to form agglomerates during sol-gel condensation [29].

Another surface characteristic of core-shell systems is the zeta potential as a measure for charge distribution at the solid-liquid interface. The zeta potential of the synthesized samples at different pH values were measured and compared against the zeta potentials of anatase and pure silica, as presented in Figure 2b. The results show that the zeta potential changes gradually towards the pure silica zeta potential with increasing synthesis steps, indicating that the surface coverage increases with the steps [30]. The isoelectric point (IEP), i.e., the intercept of the zeta potential curve with the x-axis, is the pH at which the surface carries no electric charge. The IEP can be linked to the point of zero surface charge [31–34]. This means that at pH 7 the surface of anatase, which has an IEP of 6, is slightly negatively charged by deprotonation of surface hydroxyl groups [31]. The reference sample TS0, which simulates the synthesis conditions in the absence of TEOS, shows a similar curve as the anatase, indicating that the reaction environment itself did not change the surface properties of anatase nanoparticles. The addition of silica affected the IEP considerably: The thicker the silica shell, the closer the IEP of the samples are to the IEP of pure silica (S1), which has a value of 1, where a lot of surface OH groups are dissociated. The results therefore confirm the successful tuning of surface charge density with shell formation.

### 2.1.2. Spectroscopic Measurements

The absorption properties of the core-shell structures was investigated by diffuse-reflectance spectroscopy in the range of UV to visible light. The results were converted into Tauc plots (Figure 3a), from which the optical band gaps were calculated via the intercept of the extrapolation of the tangent on the photon energy axis. The band gaps of the coated samples are similar to the one of the anatase reference (3.2 eV), indicating that the optical absorption behaviour did not change considerably.

The ATR-FTIR spectra of the coated samples are compared with the reference sample, the untreated anatase, and silica nanoparticles in Figure 3b. The two peaks, 1060 and 1180 cm$^{-1}$, visible for the samples with a shell thickness of at least one monolayer (TS2) as well as the plain silica sample

(S1), belong to the in-phase and out-of-phase asymmetric vibration of Si–O–Si, respectively [14,35,36]. The peak height increases with increasing shell thickness. The highest peak position shifts from 1055 to 1065 cm$^{-1}$ with increasing shell thickness, which indicates stronger bonds. The symmetric stretching vibration (in phase) of Si-O-Si at 790 cm$^{-1}$ is only visible for the plain silica because it is overlaid by the broad Ti-O-Ti vibration [37]. The characteristic stretching vibration of Ti-O-Si at 930 cm$^{-1}$ is visible for the coated samples and its intensity rises with increasing shell thickness until TS4 [38,39]. From these results one could conclude that the surface is completely covered with silica only for TS4 and TS5. However, the Ti-O-Si vibration is close to the vibration of Si–OH at ≈960 cm$^{-1}$, visible for the silica sample S1, which makes this conclusion debatable [35,36].

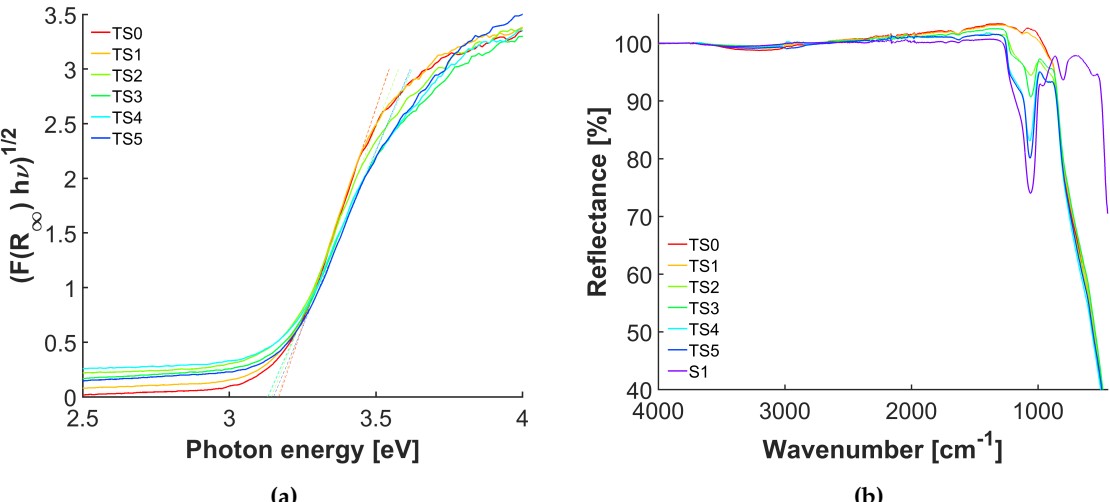

**Figure 3.** (**a**) Band gap plots of anatase and samples TS1-5; (**b**) ATR-FTIR spectra of anatase, S1, and samples TS0-5.

## 2.2. Photocatalytic Dye Degradation

We investigated the effects of silica shell thickness on the photocatalytic degradation of differently charged reactants via UV irradiation. In particular, we chose methyl orange (MO, negatively charged) and methylene blue (MB, positively charged). To monitor their degradation we analysed their decolourisation via UV-VIS, which is usually the first stage of the molecule's decomposition [40–42]. Blind measurements without a catalyst were also performed to exclude degradation by illumination alone.

The measured pH values of the catalyst suspended in dye solution were 4.4 and 4.3 for MB and MO, respectively. The titania surface is therefore slightly positively charged, while the silica surface has a negative charge.

As shown in Figure 2b, the core-shell structures exhibit IEPs that lie between the respected values of both pure oxide references. We therefore expect that methylene blue as a positively charged reactant, in contrast to methyl orange, will adsorb more strongly on the core-shell structures within increasing silica thickness.

Figure 4 shows the total adsorbed amount of both dyes on catalysts with a different shell thickness after 2 h stirring in the dark. Note that the adsorbed amount was correlated with the respective SSA$_{BET}$ (Table 1) and compared to the value of the blind measurement. The graphs show that the negatively charged reactant is indeed adsorbed to a higher extent on the positively-charged titania than the positively-charged reactant. It also shows that the adsorption capacity of the positive reactant increases and the negative one decreases with increasing shell thickness of a negatively-charged silica layer. Interestingly, with increasing silica thickness, more MO seems to adsorb. Considering that the adsorbed amounts were measured after 2 h, and thus likely at equilibrium, the increased adsorption can be attributed to the filling of micropores, which increase in content with increasing shell thickness.

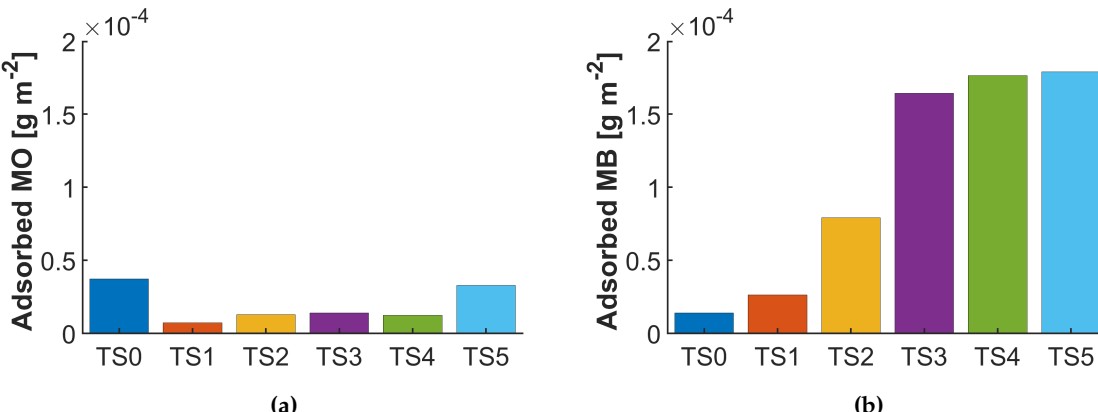

**Figure 4.** Adsorption studies of a blind and different core-shell samples of (**a**) methyl orange (MO) and the (**b**) methylene blue (MB).

Figure 5a shows the degradation in MO dye concentration upon illumination with UV light. TS1 and TS2 show a slightly increased photocatalytic performance, while the activity for thicker shells (TS3 to TS5) gradually decreases to values below that of the reference (see also Table 2). The higher activity of TS1 and TS2, compared with the reference sample TS0, is surprising, especially if we consider that the adsorption capacity of MO on these catalysts was lower than TS0 (Figure 4). This indicates that the adsorption process is not the rate limiting step in these samples. One possible explanation for this increase in activity is a better dispersion of the nanoparticle catalysts due to the repulsion of the negatively charged silica surface [25,26]. Indeed, the IEP of the coated samples would favour dispersion in neutral solutions, but is rather unlikely at a current pH of 4.4 and 4.3 for MB and MO, respectively. The increase could also be due to the reduced recombination of electron-hole pairs upon charge trapping at the silica-titania interface as discussed by Hu et al. [5]. We did not observe such changes, e.g., by solid-state photoluminescence spectroscopy. A third explanation is the formation of Lewis acid sites by Si species within the $TiO_2$ matrix at the triple-phase boundary (TPB) of support, silica islands and liquid. Ti-O-Si bonds are known to be catalytically very active and are available for the reactants on a shell thickness up to one monolayer [11,43]. Indeed, we observed an increase of Ti-O-Si in ATR-FTIR (Figure 3b) with each increasing step number. It is likely that these bonds become inaccessible beyond a monolayer of silica (i.e., >TS2). Therefore, the activity does not increase further in the thicker core-shell structures of T3 to T5, but even decreases significantly. We attribute this decrease to a charge transfer limitation, which is inhibited through the insulating layer.

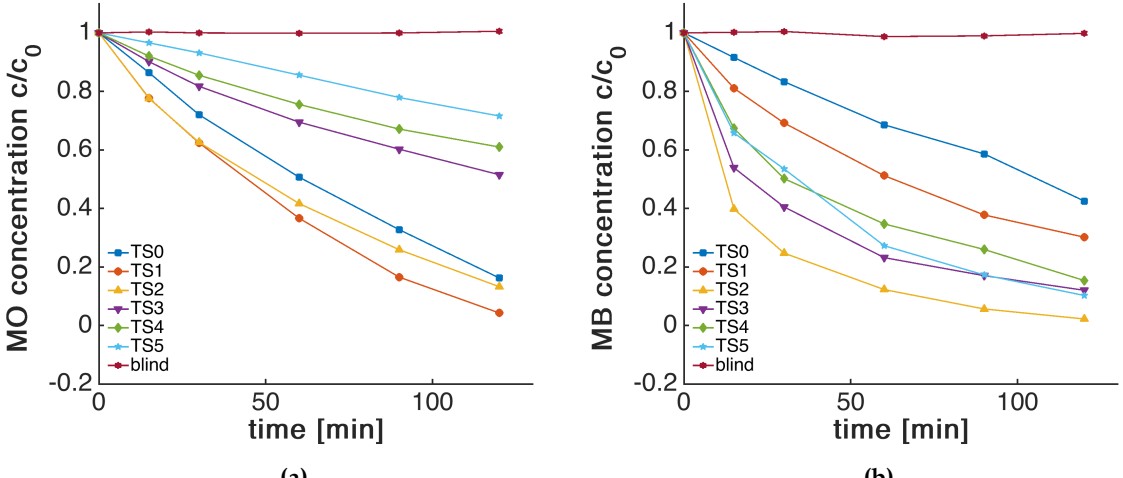

**Figure 5.** Photocatalytic degradation of a blind and different core-shell samples on the (**a**) methyl orange (MO) degradation and the (**b**) methylene blue (MB) degradation.

In the case of MB, the performance enhances strongly until TS2, beyond which it decreases again, yet still retaining an activity larger than pure titania (Figure 5b, Table 2). This behaviour can be explained by several counteracting processes. On the one hand, the increase in adsorption capacity and in accessible Lewis sites benefits the degradation process. This is offset at larger shell thicknesses, at which the charge transfer and injection into the dye reactants becomes rate limiting.

**Table 2.** $SSA_{ext.}$ normalised apparent rate constants of the first 30 min of methyl orange (MO) degradation and methylene blue (MB) degradation of different core-shell samples.

| Sample | MO rate constant $\cdot 10^{-4}$ [$s^{-1}m^{-2}$] | $R^2$ | MB rate constant $\cdot 10^{-4}$ [$s^{-1}m^{-2}$] | $R^2$ |
|---|---|---|---|---|
| TS0 | 1.95 | 0.99 | 1.09 | 0.99 |
| TS1 | 3.21 | 0.99 | 2.59 | 0.98 |
| TS2 | 4.01 | 0.99 | 13.1 | 0.99 |
| TS3 | 1.59 | 0.99 | 8.18 | 0.97 |
| TS4 | 2.03 | 0.99 | 9.21 | 0.99 |
| TS5 | 0.75 | 0.99 | 7.15 | 0.97 |

Apparent rate constants of the first 30 min of MO and MB dye degradation were determined with a pseudo-first order rate law and normalised with the $SSA_{ext.}$ calculated from the BET (Table 2). We used the $SSA_{ext.}$ to correlate our data because it is known that micropores and small mesopores impose serious kinetic limitations for pore diffusion of large hydrated reactant molecules, such as MB and MO, which is critical for dynamic processes in liquid phase, such as photocatalysis [15,44–46]. The combined results strongly suggest that below a 0.3 nm shell thickness, the adsorption, determined by surface charges, and the access to acidic Ti-O-Si sites are the rate determining factors. Beyond a thickness of 0.3 nm, the charge transfer through the insulating layer becomes the rate limiting step (Scheme 1). The mechanistic details in this work will have a great impact on the photocatalytic community, while the presented deposition process will lead to highly controllable and tunable core-shell structures for use in a wider range of applications.

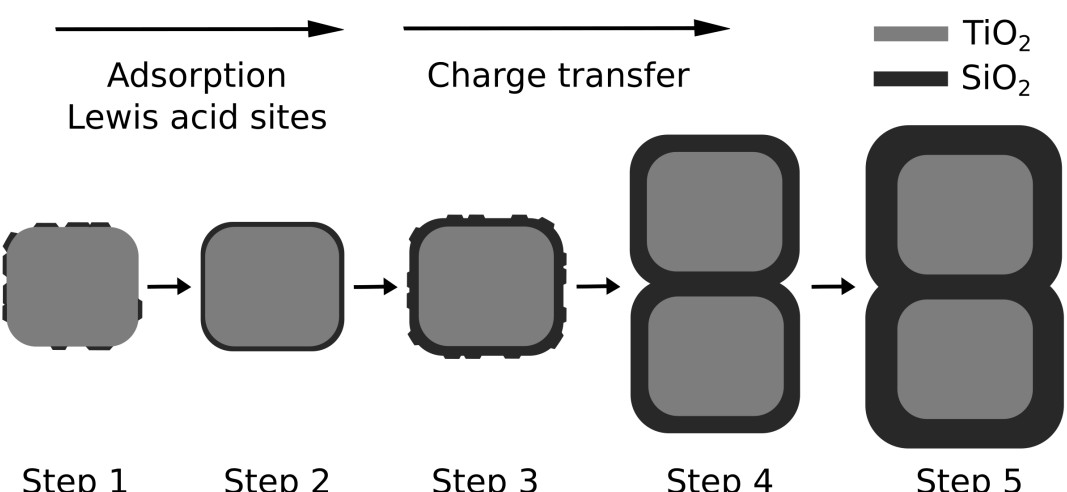

**Scheme 1.** Step-by-step growth of silica on titania during step-wise sol-gel synthesis and rate-limiting steps of dye degradation.

## 3. Materials and Methods

### 3.1. Sol-Gel Coating of $SiO_2$ onto Anatase Nanoparticles

The coating of the titania nanoparticles was performed by a layer-by-layer process based on a modified Stöber technique from the literature [47–49]. In the usual procedure, 0.25 g of $TiO_2$ anatase nanoparticles (Sigma Aldrich, 92 $m^2$/g BET surface area) were ultrasonicated in 32 mL of ethanol

(Chem-lab, Ethanol, abs., 100%) for 30 min. Then, 1.9 mL of $NH_4OH$ (27–33 wt.%) and 0.03 mL of tetraethyl orthosilicate (TEOS, 99%, ABCR) were added and the solution was stirred in a closed vessel under ambient conditions for 1 h. The steps of ultrasonication, catalyst, and precursor addition were repeated 1–5 times. After the respective final step, the material was filtered and washed with copious amounts of ethanol and hexane. It was then calcined for 1 h at 325 °C with a heating rate of 3.5 K/min. This series of samples was named TS1 to TS5, with the number hinting to the amount of steps. A reference sample was prepared by ultrasonicating the $TiO_2$ anatase nanoparticles in $NH_4OH$ in absence of TEOS (TS0). For comparison, the modified Stöber synthesis was also carried out without titania nanoparticles to get pure silica (S1).

### 3.2. Material Characterisation

Samples were dried under a vacuum at 120 °C for 360 min before $N_2$ physisorption at 77 K using a Micromeritics 2020. The specific surface areas were calculated using the Brunauer–Emmett–Teller (BET) method. Structural and elemental studies were conducted on a transmission electron microscope (TEM, Fei Tecnai F20) attached with an energy-dispersive X-Ray spectroscope (EDX, EDAX Apollo XLTW SDD). Surface characteristics were determined using a Zeta Potential Analyzer (ZetaPlus, Brookhaven). Zeta potential was calculated using the Hückel model. Spectroscopic measurements were recorded with an attenuated-total reflectance ATR -FTIR (Perkin Elmer, Spectrum Two) and a UV-VIS spectrometer equipped with a Ulbricht sphere inside a diffuse reflectance unit (Jasco V-670).

### 3.3. Photocatalytic Dye Degradation

In each experiment, 10 mg of catalyst powder was ultrasonicated for 10 min in 50 mL of an aqueous solution of methyl orange (MO, $C_{14}H_{14}N_3SO_3-Na^+$, 8 mg/L) or methylene blue (MB, $C_{16}H_{18}N_3S^+Cl^-$, 8 mg/L), and then stirred in the dark in a closed, water cooled reactor for 2 h to perform adsorption studies, where a sample of 2.5 mL was collected every 60 min. We chose a concentration that ensures an optimum in UV-VIS absorption, yet leaves the solution as diluted as possible to apply a simplified Langmuir–Hinshelwood model. The amount of catalyst was optimised to lead to a 20%–60% decolourisation for the uncoated reference $TiO_2$ catalyst, to make sure the expected changes in decolourisation due to a thicker shell are depictable. The amount of adsorbed dye was calculated by using the following equation:

$$q_t = \frac{(c_b - c_t) \cdot V}{m \cdot SSA} \tag{1}$$

where $c_t$ is the concentration of the dye in the suspension after 2 h, $c_b$ is the concentration of the solution without a catalyst after 2 h (mg/L), V is the volume of the suspension (L), m is the catalysts mass (mg), and SSA is the selective surface area of the catalyst ($m^2$/g) (Table 1). The solution was exposed to UV irradiation from a LED lamp (Thorlabs SOLIS-365C) with an emission maximum at 365 nm. The average distance between the light source and the suspension was 14 cm and the photon flux at this height was $6.7 \cdot 10^{20}$ $s^{-1}$. Aliquots of 2.5 mL were removed from the solution after intervals of irradiation filtered with a syringe filter (Polypropylene, 0.45 μm) before quantification. The decolourisation of the solutions was analysed in UV-VIS spectroscopy and the peak areas (340–600 nm for MO, 420–800 nm for MB) were evaluated to compare the photocatalytic activity of the samples.

Apparent rate constants of the first 30 min of photocatalytic decolourisation were determined with a simplified Langmuir–Hinshelwood scheme for low concentrations resembling a first-order rate law:

$$c = c_0 \cdot e^{-k \cdot t} \tag{2}$$

where $c_0$ is the initial concentration after reaching adsorption equilibrium, k is the apparent first order rate constant, and t is the time in seconds [1,50]. Graphs and function fits were generated with the software package Matlab.

## 4. Conclusions

In this work we successfully deposited ultra-thin layers of $SiO_2$ on $TiO_2$ nanoparticles with five different thicknesses in the range between 0 and 1 nm in a controlled fashion. A combined set of characterisation techniques revealed that the coatings were generally highly uniform in morphology and thickness. The core-shell structures were thoroughly characterised and evaluated towards their performance in photocatalytic dye degradation. Importantly, tests with both, positively-charged and negatively-charged dyes allowed, for the first time, a direct comparison of photocatalytic performance and the identification of rate limiting steps in this ultra-thin thickness regime.

An important observation of this work is that the shell growth seems to follow the Stranski–Krastanov model, i.e., the thinnest coating (TS1) was composed of flat islands (covering about 30% of the surface) that grew to continuous mono- and bilayers (TS2 and TS3) with a thickness of 0.29 and 0.61 nm (see Table 1), respectively, before gradually developing regular thick coatings (TS4, 0.81 nm and TS5, 0.91 nm). This observation was crucial for identifying the individual contributions to the photocatalytic performance. The presence of islands in TS1 introduced Ti-O-Si bonds at so-called triple-phase-boundaries (TPBs), which acted as active Lewis sites and were likely responsible for the observed increase in activity for both dyes. In this size regime, the accessibility of these acidic sites competed with the dye adsorption step, which could be beneficial (in the case of positively-charged MB) or without any noticeable effect (i.e., not rate-limiting, in the case of negatively-charged MO). When the shell reached a continuous monolayer (TS2), these TPB sites were not available anymore, instead the effect of adsorption became dominating, which led to a stark increase in activity for the positively-charged MB and a decrease for MO. Finally, at thicker coatings (TS3 and higher), dye adsorption began to compete with charge transport through the insulating shell, which became rate-limiting.

Our work demonstrates that the performance of a photocatalyst can be tuned through the deposition of ultra-thin shells, with the best performance being defined by a trade-off between the accessibility of TPBs and the optimum strength of dye adsorption. The beneficial contribution of TBPs is a surprising observation that needs to be investigated further, in particular with respect to unravelling reaction mechanisms. This work will thus impact future fundamental photocatalytic research as well as advance the design of functional core-shell materials for a wide range of applications with similar rate-limiting regimes.

**Author Contributions:** Conceptualisation and methodology D.E. and A.G.; investigation J.B., S.N. and A.G.; visualisation A.G.; resources, project administration, funding acquisition D.E.

**Funding:** Open Access Funding by TU Wien.

**Acknowledgments:** TEM and EDX analysis was carried out using facilities at the University Service Centre for Transmission Electron Microscopy, Vienna University of Technology, Austria.

**Conflicts of Interest:** The authors declare no conflict of interest.

## Abbreviations

The following abbreviations are used in this manuscript:

| | |
|---|---|
| ATR-FTIR | Attenuated Total Reflection Fourier-Transform Infrared Spectroscopy |
| BET | Brunauer-Emmett-Teller |
| EDX | Energy Dispersive X-Ray Spectroscopy |
| MB | Methylene Blue |
| MO | Methyl Orange |
| IEP | Isoelectric Point |
| SA | Surface Area |
| SSA | Selective Surface Area |
| TEM | Transmission Electron Microscopy |
| TEOS | Tetraethyl Orthosilicate |

TPB    Tripe Phase Boundary
US      Ultrasonication
UV      Ultraviolet

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
