# Peer review of "Rate-Limiting Steps of Dye Degradation over Titania-Silica Core-Shell Photocatalysts"

_catalysts, doi:10.3390/catal9070583_

Round 1
Reviewer 1 Report
General Points:
The authors present the preparation of Ti-O-Si layers between zero and 1 nm and follow their growth and composition.They present some evidence for: a) the rate limiting step below one monolayer (0.2 nm) relates to the dye adsorption determined by the charge of the surface and the interactions with the Ti-O-S iduring the photocatalytic degradation of positive and negative dyes and b) the charge transfer limitations for thin layers around > 1 nm.This subject has been investigated very scarcely in the past. The point is well taken.
The authors develop a layer by layer process for the Ti-O-Si layer deposition. The change in the Z-potential of the Ti-O-Si particles indicate that the silica is located in the outer particle shell
The experiments are clearly described and seem feasible and the results reported possible.The techniques used are adequate and up to date: EM,EDS, Z-potential, DRS, FTIR, BET.
Specific points
Abstract: Delete first 2 sentences. They belong in the section Introduction.
line 43: thinner coating ,Thinner than what?...complete the sentence
line 77:inner surface within micropores. What diameter you assign to the micropores, generally in TiO2 they are quoted at 2 nm
Please cite the paper: T. Yuranova,R. Mosteo.... Cotton Textiles Modified by SiO2/TiO2 with Self-Cleaning Properties, 2005, 244, 160-167.
line 139: Delete "transient charges" and note instead "degradation"
Author Response
We thank the referee for this very positive and highly encouraging assessment of the novelty and impact of our work. We also thank for the specific suggestions for corrections.
Page 1: We deleted the first two sentences from the abstract and included the first sentence at a suitable position in the introduction section (line 28-29)
Page 2, lines 45-46: We specified the thickness of the coating to read “coating thinner than 2 nm” and moved the corresponding references “[17,24–26]” at the end of this sentence.
3. Page 3, line 89: We added the text “(pores with a diameter < 2 nm)” with Reference 17 to specify “micropores”
4. Page 6, line 152: We replaced “transient changes” with “degradation”
Reviewer 2 Report
The paper of Giesriegl and co-workers deals with the preparation, the structural characterization of ultra-thin layers of silica on anatase nanoparticles prepared by means of layer-by-layer sol-gel process and photocatalysts activity evaluated towards adsorption and degradation of dyes under UV illumination. TEM, EDX, ATR-FTIR, Physisorption studies and zeta potential data were provided for the characterization of the material. As for the photocatalysts activity adsorption and degradation of the catalysts towards positively-charged and negatively charged dyes were reported.
The subject topic is interesting and may data if presented in a different way could be valuable in this field. However, there are important flaws and weakness in the form of the manuscript which is prepared in a confused way. The authors claims that …”we developed a layer-by-layer sol-gel process that allowed for sophisticated deposition of ultra-thin layers of silica…” but the novelty of this method is not obvious or at least clear.. Several other authors used very similar approach 10.1021/la703899j. Thus the authors must (much) better explain and demonstrate the novelty of the lBl procedure. The introduction is to short and does not explain connection with recent development in the field. The same for the discussion section. No conclusion section were provided. The methods section should be completely reworked. It is not clear to me if the entire reaction was performed with open or closed recipient. If the recipient was left open. Did the authors take into account the evaporation extent…and so on. When the authors refers to degradation are they sure the dyes were degraded or may they refer to decolourization process (MS measurements)? Did the authors considered the role of Oxygen? Minor points, figure 1 is reported in the introduction section …resolution of figure 1 is to low. Methods section should be reported before. The organization of the manuscript does not follow the Journal guideline.
The main criticism: generally the manuscript should have significant advance in understanding, and be understandable within the framework of existing knowledge. The novelty of the presented data should be unequivocally clear. The manuscript failed to do this convincingly. For these reasons in my opinion, the manuscript cannot be accepted according to the reported reasons.
Author Response
The paper of Giesriegl and co-workers deals with the preparation, the structural characterization of ultra-thin layers of silica on anatase nanoparticles prepared by means of layer-by-layer sol-gel process and photocatalysts activity evaluated towards adsorption and degradation of dyes under UV illumination. TEM, EDX, ATR-FTIR, Physisorption studies and zeta potential data were provided for the characterization of the material. As for the photocatalysts activity adsorption and degradation of the catalysts towards positively-charged and negatively charged dyes were reported.
The subject topic is interesting and may data if presented in a different way could be valuable in this field.
We thank the referee for this positive assessment. We understand the referee’s concern regarding the didactical presentation of data and explanation of novelty and thank the referee particularly for the detailed suggestions for improvements, which we have addressed with great care.
The authors claims that …”we developed a layer-by-layer sol-gel process that allowed for sophisticated deposition of ultra-thin layers of silica…” but the novelty of this method is not obvious or at least clear.. Several other authors used very similar approach 10.1021/la703899j. Thus the authors must (much) better explain and demonstrate the novelty of the lBl procedure.
The referee is right in this assessment: we did not develop the coating process, but adjusted one to our specific needs. We understand that such message may have been conveyed. We revised the wording on page 2, line 62 (replaced “developed” by “synthesized”) and on page 7, lines 189-190, to read: “performed by an extended layer-by-layer process based on a modified Stöber technique from literature”, where we also added the corresponding, new references [47-49], including the referee’s suggestion.
The introduction is to short and does not explain connection with recent development in the field. generally the manuscript should have significant advance in understanding, and be understandable within the framework of existing knowledge. The novelty of the presented data should be unequivocally clear.
In this particular field of core-shell structures with ultrathin coatings (thinner than 2 nm) for application in photocatalysis, not much has been reported; hence it is understandable that the introduction section is not longer and might even appear short; yet it is rather concise and includes the important references. Nevertheless, we revised the introduction section and added some additional references to clarify both, novelty and impact:
Page 1, lines 28ff: We added the following text and references: “Core-shell structures can offer a tunable model system for functional and mechanistic investigations in heterogeneous catalysis. [9,10] Typically, the core-shell materials are inorganic, most often including SiO2 [9,10]. As an inter material, it is either used as support for a catalytic shell or as an insulating shell around a catalyst [2,5,6,11-13]. As observed in literature the coating of semiconduactor photocatalysts with SiO2 can have several advantages.”
Page 2, lines 45-47: We specified: “So far, only a few groups have investigated the effects of a thinner coating coating thinner than 1 nm, often with irregular thickness and morphology or without comparing the photocatalytic effects on differently charged reactants. [17,24–26].”
Page 1, lines 53-58: We revised to read: “Therefore, in this work we prepared continuous films with uniform thickness adjusted between 0 and 1 nm. We compared, for the first time, the photocatalytic effects on differently charged dyes in this crucial thickness range. This work has allowed to unravel charge transfer limitations, reveal differences in adsorption behaviour and accessibility to acidic sites and, thus, to identify rate limiting steps for the respective thickness regimes, which is crucial for the community to further advance the design of photocatalysts.”
We furthermore added a detailed conclusion and revised the discussion (see below). Together with the revisions in the introduction, we believe that the novelty and, in particular, the impact of this work is better highlighted in the revised manuscript.
No conclusion section were provided.
We agree with the referee that a concluding section would significantly enhance this manuscript. In the previous version, we decided to omit it in order to conform to the Journal’s recommendations. After corresponding with the Editor, we agreed to add this important section (page 8, lines 237 ff):
“In this work we successfully deposited ultrathin layers of SiO2 on TiO2 nanoparticles with 5 different thicknesses in the range between 0 and 1 nm in a controlled fashion. A combined set of characterisation techniques revealed that the coatings were generally highly uniform in morphology and thickness.
The core-shell structures were thoroughly characterized and evaluated towards their performance in photocatalytic dye degradation. Importantly, tests with both, positively-charged and negatively-charged dyes, allowed, for the first time, a direct comparison of photocatalytic performance and the identification of rate limiting steps in this ultrathin thickness regime.
An important observation of this work is that the shell growth seems to follow the Stranski-Krastanov model, i.e. the thinnest coating (TS1) is composed of flat islands (covering about 30% of the surface) that grow to continuous mono- and bilayers (TS2 and TS3) with a thickness of 0.29 and 0.61 nm (see Table x), respectively, before gradually developing regular thick coatings (TS4, 0.81 nm and TS5, 0.91 nm). This observation is crucial for identifying the individual contributions to the photocatalytic performance. The presence of islands in TS1 introduces Ti-O-Si bonds at so-called triple-phase-boundaries (TPBs), which act as active Lewis sites and are likely responsible for the observed increase in activity for both dyes. In this size regime the accessibility of these acidic sites competes with the dye adsorption step, which can be beneficial (in the case of positively charged MB) or without any noticeable effect (i.e. not rate-limiting, in the case of negatively-charged MO). When the shell reaches a continuous monolayer (TS2), these TPB sites are not available anymore, instead the effect of adsorption becomes dominating; this leads to a stark increase in activity for the positively charged MB and a decrease for MO. Finally, at thicker coatings (TS3 and higher), dye adsorption begins to compete with charge transport through the insulating shell, which now becomes rate-limiting.
Our work demonstrates that the performance of a photocatalyst can be tuned through the deposition of ultrathin shells, with the best performance being defined by a trade-off between the accessibility of TPBs and the optimum strength of dye adsorption. The beneficial contribution of TBPs is a surprising observation that needs to be investigated further, in particular with respect to unravelling reaction mechanisms. This work will thus impact on future fundamental photocatalytic research as well as advance the design of functional core-shell materials for a wide range of applications with similar rate-limiting regimes.”
The methods section should be completely reworked. Methods section should be reported before. The organization of the manuscript does not follow the Journal guideline.
The current placement of the Methods sections is conform with the Journal’s guidelines. Indeed, we would also prefer to move it to the more traditional position between Introduction and Results sections.
It is not clear to me if the entire reaction was performed with open or closed recipient. If the recipient was left open. Did the authors take into account the evaporation extent?
We thank the referee for these important comments.
The synthesis was indeed performed in a closed vessel. We added this information on page 8, line 194 to read: “the solution was stirred in a closed vessel under ambient conditions for one hour”
We also clarified the synthesis of the reference sample in lines 199-201.
Also, the photocatalytic dye degradation was also performed in a closed reactor. We added this information on page 8, line 214. We also added a short paragraph to describe the choice of dye concentration in the reaction solution (page 8, lines 215 to 219).
When the authors refers to degradation are they sure the dyes were degraded or may they refer to decolourization process (MS measurements)?
Indeed, we measured the decolorization in the visible range, which, according to MS studies (10.1016/j.apcata.2015.08.036, 10.1021/ie403402q) is usually followed by several mechanistic steps of degradation/mineralization. According to 10.1016/j.saa.2015.05.067, the degradation of aromatic structures can be monitored by absorption in the UV range. Our UV-VIS measurements show, that both processes are accompanied. To clarify which process we analyzed, we changed “degradation” to “decolorization” in the methods section. We also explained the choice for measuring decolorization on page 6, line 131-132 and changed the term “degradation rates” to “apparent degradation rates”
Did the authors considered the role of Oxygen?
This is a very good question, often overlooked in literature. For the response, we estimated the amount of oxygen available to the reactant (both dissolved and in atmosphere above solution) and compared it to the maximum amount required for total mineralization of organic products (we would need far less than that):
Our reactor has a total volume of 300 ml, i.e. 250 ml of air and 50 ml of dye solution. At the temperature of the cooled reactor, e.g. below 10 °C, about 0.18 *10-4 mol (18 mmol) O2 is dissolved in the solution, while 2.3*10-5 mol (2.3 mmol) O2 remains in the gas phase of the closed reactor. The amount of dye in our 50 ml solution is about 1.25 µmol, i.e. the corresponding amount of carbon atoms (16 for MB) amasses to 20 µmole. Thus, for total mineralization to CO2 we need 20 µmole of O2, which is close to amount of dissolved O2. If we further consider, that O2 is constantly resupplied from the air to achieve equilibrium, there is enough O2 available in the reactor to not impose a kinetic restriction.
Minor points, figure 1 is reported in the introduction section …resolution of figure 1 is to low.
We have moved Figure 1 to the Results section. The images were taken at the maximum resolution possible with Gatan Digital Micrograph (4k).
Reviewer 3 Report
The selection of active photocatalysts able to lead to contaminants removal through photocatalytic oxidation is an interesting theme. The paper is well written and presents interesting results. It can be published after minor revisions:
More quantitative data must be given in abstract
Novelty and objectives of the paper must be highlighted
Have the authors tested the catalyst reutilization and stability?
Was the catalysts preparation methodology previously optimized? If so, some references must be given
The concentration of catalyst and dyes applied must be justified.
Author Response
The selection of active photocatalysts able to lead to contaminants removal through photocatalytic oxidation is an interesting theme. The paper is well written and presents interesting results. It can be published after minor revisions:
We thank the referee for this very positive and highly encouraging assessment of the novelty and impact of our work. We also thank for the specific suggestions for corrections.
More quantitative data must be given in abstract
Abstract: We revised the abstract and added more quantitative data:
In this work, we employed a step-by-step sol-gel process to controllably deposit ultra-thin layers of silica on anatase nanoparticles in the range between 0 and 1nm. The deposition was confirmed by TEM, EDX, ATR-FTIR (e.g. Ti-O-Si band at 960 cm–1). Zeta potential measurements unravelled a continuous change in surface charge density with increasing silica shell thickness. The photocatalysts were evaluated towards adsorption and degradation of positively-charged and negatively charged dyes (methylene blue, methyl orange) under UV illumination.
The growth mechanism follows the Stranski-Krastanov model with three thickness regimes: a) flat islands (1st step), b) mono/bilayers (2nd /3rd step), and c) regular thick films (4th /5th step). The results suggest different rate limiting processes for these regimes: a) for the thinnest scenario, acidic triple-phase boundaries (TPBs) increase the activity for both dyes with their accessibility being the rate limiting step; b) for continuous mono/bilayers, dye adsorption on the negatively charged SiO2 shells becomes the rate liming step; this leads to a stark increase in activity for the positively charged MB and a decrease for MO; c) for thicker shells, the activity decreases for both dyes and is limited by the charge transport through the isolating shells. This work will benefit future fundamental photocatalytic research and advance the design of functional core-shell materials for a applications with similar rate-limiting regimes.
Novelty and objectives of the paper must be highlighted
We revised abstract and introduction section and added a conclusion section, which should better highlight the novelty and, in particular, the impact of this work in the revised manuscript.
Have the authors tested the catalyst reutilization and stability?
The chosen sample amounts were too small to test reusability. But it is a very good idea and we will perform such tests, once we succeed to scale-up these catalysts while preserving the nature and uniformity of the shells.
Was the catalysts preparation methodology previously optimized? If so, some references must be given
We modified an existing procedure adjusted to our specific needs. To clarify this, we revised the wording on page 2, line 62 (replaced “developed” by “synthesized”) and on page 7, lines 189-190, to read: “We performed by an extended layer-by-layer process based on a modified Stöber technique from literature”, where we also added the corresponding references [47-49].
The concentration of catalyst and dyes applied must be justified.
We thank the referee for this valuable comment. Indeed, we have performed several preliminary tests with various dye concentrations. At the end, we chose a concentration that ensures an optimum in UV-VIS absorption, yet leaves the solution as diluted as possible to apply a simplified Langmuir-Hinshelwood model. The amount of catalyst was chosen so that the reference (uncoated TiO2) would lead to a dye discoloration of 20-50 % at a given reaction time, to make sure the expected changes in discoloration (less for MO, more for MB) are observable, even with a thicker shell. We added a small paragraph in the methods section (page 8, line 215-219).